# HIV-1-Mediated Acceleration of Oncovirus-Related Non-AIDS-Defining Cancers

**DOI:** 10.3390/biomedicines10040768

**Published:** 2022-03-25

**Authors:** Jessica Proulx, Maria Ghaly, In-Woo Park, Kathleen Borgmann

**Affiliations:** Department of Microbiology, Immunology and Genetics, University of North Texas Health Science Center, Fort Worth, TX 76107, USA; jessicaproulx@my.unthsc.edu (J.P.); mariaghaly@utexas.edu (M.G.); inwoo.park@unthsc.edu (I.-W.P.)

**Keywords:** human immunodeficiency virus, hepatitis, HBV, HCV, human papillomavirus, HPV

## Abstract

With the advent of combination antiretroviral therapy (cART), overall survival has been improved, and the incidence of acquired immunodeficiency syndrome (AIDS)-defining cancers has also been remarkably reduced. However, non-AIDS-defining cancers among human immunodeficiency virus-1 (HIV-1)-associated malignancies have increased significantly so that cancer is the leading cause of death in people living with HIV in certain highly developed countries, such as France. However, it is currently unknown how HIV-1 infection raises oncogenic virus-mediated cancer risks in the HIV-1 and oncogenic virus co-infected patients, and thus elucidation of the molecular mechanisms for how HIV-1 expedites the oncogenic viruses-triggered tumorigenesis in the co-infected hosts is imperative for developing therapeutics to cure or impede the carcinogenesis. Hence, this review is focused on HIV-1 and oncogenic virus co-infection-mediated molecular processes in the acceleration of non-AIDS-defining cancers.

## 1. Introduction

Human immunodeficiency virus-1 (HIV-1) infection accelerates the progression of both acquired immunodeficiency syndrome (AIDS)-defining cancers and non-AIDS-defining cancers in patients co-infected with HIV-1 and other oncogenic viruses. With the advent of combination antiretroviral therapy (cART), AIDS-defining cancers, such as Kaposi sarcoma and non-Hodgkin lymphoma have declined by 60–70% and 30–50%, respectively, when compared with the pre-cART era [1]. Significant reduction of the rates of these cancers by cART is believed to be due to ameliorating the suppressed immune responses, including depletion of CD4+ T helper cells, exhaustion of lymphopoiesis, and so forth [2,3,4,5]. By contrast, non-AIDS-defining cancers caused by hepatitis B and C viruses (HBV and HCV), human papillomavirus (HPV), etc., in HIV-1-associated malignancies have been significantly elevated [6,7,8] such that cancer is the leading cause of death in HIV-1-infected persons in certain highly developed countries [9], where the life expectancy of HIV-1-infected people has increased with the advent of cART. However, it is currently unknown how HIV-1 raises the risks for non-AIDS-defining cancers; thus, understanding is critical for developing therapeutics to cure or impede the carcinogenesis.

Currently, cART is composed of six different classes of antiretroviral drugs, such as entry/attachment inhibitors, nucleoside and nonnucleoside reverse transcriptase inhibitors, protease/integrase inhibitors, and so forth [10,11]. cART blocks HIV-1 replication at different stages of the HIV-1 life cycle and significantly reduces HIV-1 infection-associated morbidity and mortality [12,13,14,15]. Blockade of HIV-1 replication with cART reduces the viral load and thus the risk of HIV-1 transmission. Moreover, the treatment allows the immune system destroyed by HIV-1 infection to repair and prevent further damage [16,17,18]. As virus replication is blocked and thus immune responses are recovered, risks of AIDS-defining cancers are remarkably reduced, while those of non-AIDS-defining cancers were increased, as briefly described above. However, it is currently unknown how cART differentially impacts the progression of either AIDS- or non-AIDS-defining cancers. This review is focused on the co-infection-associated impacts on tumorigenesis of non-AIDS-defining cancers by HIV-1 infection, illustrating potential intercommunication between HIV-1 and oncoviruses in the co-infected hosts to develop these cancers.

## 2. Overview of HIV-1 and Oncoviruses

### 2.1. HIV-1

It is unclear how HIV-1 infection leads to clinical deterioration in oncovirus-caused cancers, even if HIV-1 and the co-infected oncoviruses have disparate routes of transmission and target cells for infection. First, HIV-1 is transmitted by sexual contact across mucosal surfaces, percutaneous inoculation, or by maternal–infant exposure [19]. These transmission mechanisms differ from those of oncoviruses with the exception of HBV and HCV. Transmitted HIV-1 in the human body then enters the target cell as the first step of the virus life cycle, by the interaction between HIV-1 envelope (Env) and the receptor/co-receptor molecules expressed on the surface of the host cells. Briefly, Env is composed of a glycoprotein (gp) 120 trimer on the surface of the virion particle and heterodimer of gp41 in the virus membrane. HIV-1 gp120 first binds to its primary receptor, a cluster of differentiation 4 (CD4) expressed on the surface of the target cells [20,21,22,23,24]. Interaction between gp120 and CD4 is critical for the repositioning of variable loop 3 of gp120 to enhance co-receptor engagement. Different HIV-1 strains employ disparate co-receptor molecules for virus entry. For instance, HIV-1 strains that target CD4+ T lymphocytes utilizes the C-X-C motif chemokine receptor 4 (CXCR4) as a co-receptor (X4-tropic or T-tropic), the virus entering into macrophage/monocytes and dendritic cells uses the C-C motif chemokine receptor 5 (CCR5) as a co-receptor (R5-tropic or M-tropic), and primary isolates employ both CXCR4 and CCR5 for virus entry (R5X4-tropic or dual tropic) [25,26,27,28,29]. In virally suppressed individuals, the largest HIV reservoir resides in various subsets of memory T cells and the importance of tissue reservoirs remains highly discussed [30,31,32]. The most compelling evidence confirms that HIV-1 enters its target cells using CCR5 and CXCR4, indicating that in vivo HIV-1 targets cells expressing CD4 and CXCR4 and/or CCR5 [27]. That is, HIV-1 susceptible cells are distinct from those for oncoviruses, as described below (Table 1).

While co-infection of cells by both an oncovirus and HIV-1 is controversial, as described below, co-infection may not be the only way for HIV-1 to accelerate oncovirus-mediated tumorigenesis. In fact, transferable HIV-1 proteins may expedite oncovirus-mediated cancer progression (Figure 1). Briefly, HIV-1 expresses several important regulatory proteins (Tat, Rev, Nef, Vpr) in addition to three structural proteins (Gag, Pol, and Env), in a stage-specific manner. Regulatory proteins such as transactivator of transcription (Tat), regulator of expression of virion proteins (Rev), and negative factor (Nef) are expressed during the early stage (Figure 2). More specifically, Rev exports viral messages containing the Rev-response element to the cytoplasm so that structural polyproteins encoded by gag, pol, and env, are produced at the late stage of virus infection [33,34,35,36,37,38]. Among these viral proteins, Env, Tat, and Nef are key viral proteins that can be transferred from HIV-1-infected cells, by means other than infectious virions, into target cells and dysregulate the target cell biology. Since HIV-1 Tat is a diffusible, cell-penetrating peptide [39], secreted Tat could diffuse into neighboring cells and modulate intracellular gene expression, tumorigenesis, etc. [40,41]. Furthermore, the HIV envelope gp120 could directly interact with CXCR4 or CCR5 co-receptor molecules expressed on the non-CD4+ target cells [42,43], dysregulating the co-receptor-mediated intracellular signaling cascade and changing the target cell biology [44,45]. HIV-1 Nef can also be transferred from HIV-1-infected cells to the non-susceptible cells through conduits (filopodia) and/or exosomes [46,47,48,49]. However, it is unclear how these HIV-1 proteins contribute to the escalation of non-AIDS-defining cancers in the co-infected patients.

#### Co-Infection with Oncoviruses

Among non-AIDS-defining cancers, this review is focused on the oncoviruses that most commonly affect people living with HIV. Due to the shared route of transmission between HIV-1 and HCV, an estimated 15 to 30% of persons living with HIV-1 have or have had an HCV co-infection [50,51,52] in the US. While both HBV and HCV are treatable, chronic liver disease remains a major cause of morbidity and mortality in people living with HIV. In people living with HPV, an HIV-1 co-morbidity increases the risk for head and neck squamous cell carcinoma, anal cancer, and cervical cancer between 2- and 4-fold [8,53,54]. This review is focused only on oncogenic virus-mediated non-AIDS-defining cancers, excluding AIDS-defining cancers, such as Epstein–Barr virus and human herpesvirus. 

### 2.2. Hepatitis B and C Viruses (HBV and HCV)

Transmission and entry. HBV and HCV are responsible for the most widespread forms of chronic hepatitis worldwide and lead to liver disease, cirrhosis, and hepatocellular carcinoma (HCC) [55,56,57,58,59,60,61]. Like HIV, they are transmitted by sexual contact, sharing needles, syringes, and other injection equipment, and by maternal–infant exposure. HBV and HCV enter hepatocytes by interacting with distinctively disparate receptor molecules. HBV enters cells by interaction of the preS1 region of the large (L) HBV glycoprotein with a liver-specific bile acid transporter known as the sodium taurocholate co-transporting polypeptide (NTCP) [62,63,64,65,66,67,68]. However, a recent report indicates that NTCP itself is not sufficient for entry, and E-cadherin plays a role in HBV entry by modulating the localization of NTCP to the plasma membrane [69]. It is also reported that epidermal growth factor receptor enhances HBV internalization by complex formation with NTCP [70]. By contrast, HCV entry requires multiple receptors, including tetraspanin CD81 [71], scavenger receptor class B type I (SR-B1) [72], tight junction proteins, claudin-1 (CLDN1) [73], and occludin (OCLN) [74]. Despite differential routes of entry, both hepatitis viruses cause morbidity and mortality through the development of chronic liver diseases, including cirrhosis and HCC [58,59,60,61]. 

Life cycle. The life cycles of HBV and HCV are as distinct as their receptors despite inducing similar pathogenesis. As a member of the Hepadnaviridae family, HBV’s life cycle includes the nucleus where its partially double-stranded (ds) DNA genome is integrated into the cell genome [75]. While, HCV, as a member of the Flaviviridae family, possesses an RNA genome and completes its life cycle in the cytoplasm. 

The HBV viral genome is transported to the nucleus wherein a covalently closed circular form of DNA (cccDNA) accumulates as a stable episome, and the cccDNA is responsible for persistent HBV infection of hepatocytes, serving as a template for viral transcription [76,77,78,79]. Subgenomic and pregenomic RNA transcribed from the template then function as a template for reverse transcriptase as well as mRNA to generate viral proteins, such as core, pre-core, polymerase, small (S), middle (M), and large (L) glycoproteins, and the critical HBV-encoded regulatory protein hepatitis B virus X protein (HBx) (Figure 3). The translated viral proteins are assembled in the endoplasmic reticulum (ER) and Golgi to generate viral particles [76]. The pregenomic RNA is packaged into the core and reverse transcribed into (−) strand DNA by viral reverse transcriptase, which serves as a template for (+) strand DNA, forming a partially ds-genomic DNA. The assembled HBV particles are then released from the hepatocyte ER [76] in vesicles that travel to the plasma membrane. 

In contrast, HCV, the Hepacivirus genus of the Flaviviridae family, comprises a single stranded (ss) RNA genome with messenger activity in the virus particle. Since the genomic RNA is translated into a single polyprotein precursor upon entry, the virus spends its entire lifecycle in the cytoplasm. The polyprotein is then cleaved into several smaller structural (core, envelope 1 and 2) and non-structural (NS1 to NS5A and 5B) elements (Figure 4). The virus does not have a reverse transcriptase protein and thus does not integrate into the host genome like HBV. However, the viral core proteins (p19 and p21) enter the nucleus and trigger various cellular signal transduction pathways to mediate the transcription activity of nuclear factor kappa B (NF-κB) and signal transducer and activator of transcription (STAT) 3 proteins. It is known that HCV infection dysregulates the host immune responses of infected cells, inducing hepatitis. 

#### 2.2.1. Molecular Mechanisms on HBV- and HCV-Mediated Hepatocarcinogenesis

As noted above, HBV and HCV use different receptor molecules to enter hepatocytes and have distinct virus life cycles within the cells. Nonetheless, the eventual consequence can be identical, HCC. 

HBV replicability, genotype, and genomic mutations. HBV, directly and indirectly, dysregulates host cell biology in various manners to induce HCC. First, the risk of developing HCC is directly related to replicability, genotype, and genomic mutations of HBV [58,80]. High serum HBV DNA generated and released during viral replication is an indicator of accumulation of liver damage, evolution to cirrhosis, and HCC development [58]. Among the eight HBV genotypes (A–H), genotype C has been associated with a higher risk of progression to HCC [81]. Mutations in the basal core promoter of HBV genotypes B and C are closely associated with the development of HCC. More specifically, deletions, insertions, or even nonsynonymous point mutations in the preS1 and preS2 gene found in patients with cirrhosis and long-lasting HCC [82,83,84,85,86,87,88] increased the risk of HCC by 3.77-fold [83]. These risks are the result of mutant protein production that abnormally activates the ER stress signaling pathways [82,89,90,91], generating reactive oxygen species (ROS) and causing oxidative DNA damage and genomic instability [92,93,94]. Such changes in the different strains of HBV provide an optimal microenvironment for the development of HCC.

HBV DNA integration. The integration of HBV DNA into the host genome during the early steps of clonal tumor expansion enhances genomic instability and mutagenesis and causes secondary chromosomal rearrangements, such as duplications, translocations, and deletions, in a diverse array of cancer-related genes. Genome-wide association studies indicated that single nucleotide polymorphisms in genes such as deleted in liver cancer 1 (*DLC1*, a tumor suppressor gene implicated in HCC pathogenesis) [95], *STAT4* (a key protein for regulation of the inflammation) [96,97], cytotoxic T-lymphocyte antigen (*CTL*)-4 [97], transforming growth factor β 1 (*TGF*-β1) [98], *TPTE2* (a phosphatase and tensin homolog) [99], mouse double minute 2 homolog (*MDM2*) [100] are associated with HCC in chronic HBV patients. This demonstrates that genetic mutation by HBV infection plays a pivotal role in HCC development. Further, somatic gene mutations in the coding region in HCC [101,102,103,104,105,106], reactivation of telomerase reverse transcriptase (*TERT*), which is important for telomerase maintenance, and promoter mutations caused by HBV insertion and/or HBx protein [101,102,103,106] are frequently observed in HCC, indicating that various mutations contribute to induction and acceleration of HBV-mediated tumorigenesis. 

HBV gene expression. Aberrant expression and/or functional changes of HBV viral genes and proteins trigger HBV-associated tumorigenesis by dysregulating the expression of viral and cellular genes integral to cell functions, such as cell proliferation and viability, and sensitizing liver cells to mutagens. For instance, HBx plays a critical role in cell proliferation and death by regulating the expression of various cellular genes involved in ROS production [107], ER stress [108], apoptosis [109], angiogenesis [110,111], etc., and viral transcription of the large S gene, which promotes hepatocarcinogenesis [112,113]. To elaborate, HBx augments HBV replication by recruiting the cccDNA minichromosome in HBV replicating cells [114,115] and by blocking the inhibitory activity of HBV transcription conferred by protein arginine N-methyltransferase 1 (PRMT1) [116,117], spindlin-1 [117], and SET domain-bifurcated histone lysine methyltransferase 1 (SETDB1) [118]. Regulation of HBV replication is also mediated by additional mechanisms, such as transcriptional activation of genes and micro (mi)RNAs potentiating endocytosis and autophagy [119,120,121], inactivation of the Smc5/Smc6-associated restriction of cccDNA transcription [122], and direct transcriptional repression of miRNAs (miR-138, -224, and -596) that inhibit HBV replication [119]. The HBV capsid protein is also known to bind to the viral minichromosome [77,79], affect cccDNA transcription [79] and stability, and regulate the expression of host genes involved in innate immunity, inflammation, and cell proliferation [123]. Inappropriate expression of wild-type envelope initiates a cascade of events that advance malignant transformation [112]. Moreover, mutated preS2 displays pro-oncogenic potential via transcriptional activation of TERT [91,124], upregulation of cyclin A [125], and enhancement of vascular endothelial growth factor-A [126]. Mutant preS2 also dysregulates phosphorylation of retinoblastoma (Rb) and cell cycle progression [127], induces cell proliferation and anchorage-independent growth by overexpression of cyclin A and cyclooxygenase-2 [93,128], and thereby promotes HCC development. Taken together, changes to the viral gene expression and functional changes of viral genes by mutation play a pivotal role in regulation of HCC progression.

Prolonged expression of viral genes in infected cells induces mutations in various cellular genes involved in the signal transduction pathways critical to tumorigenesis. In HCC, the WNT/β-catenin pathway is activated by activating and inactivating mutations of *CTNNB1* [129,130] and *AXIN1* [130], respectively. Approximately 5–15% of HCC involve mutations that activate nuclear factor-erythroid factor 2-related factor 2 (NRF2) and inactivate Kelch-like ECH-associated protein 1 (KEAP1), which prevents NRF2 ubiquitination and subsequent degradation. These mutations protect tumor cells from ROS-induced death by increasing antioxidant responses [103,104,131,132]. The cell cycle regulating pathway is also disrupted by mutations in p53 tumor suppressor protein in 12–48% of cases of HBV-associated HCC [102,103,131,133], which are associated with tumor aggressiveness and poor prognosis [102,105]. The Rb pathway, which plays an essential role in G1/S cell cycle regulation, is also inactivated by homozygous deletion of cyclin-dependent kinase inhibitor 2A (*CDKN2A*, CDK2) [103], amplification of the *CCND1/FGF19* locus [134,135], and recurrent HBV insertions in cyclin E1 (*CCNE1*) [106]. The fate of infected cells, whether proliferation or death, is also regulated by HBx-triggered signaling pathways. For example, HBx-mediated calcium signaling leads to the accumulation of cytosolic calcium by modulating mitochondrial calcium uptake, which in turn increases ROS, ER stress and activates the unfolded protein response pathways [108,136]. In fact, the binding of HBx to the mitochondrial voltage-dependent anion-selective channel protein 3 (VDAC3) augments ROS production, which eventually leads to cell death by apoptosis [109]. An additional effect of HBx is inhibition of tumor necrosis factor-α (TNFα) and Fas-mediated apoptosis by activation of NF-κB, enabling HBV-infected hepatocyte survival, but also uninfected cell apoptosis [137]. Thus, HBx confers differential impacts on the fates of HBV-infected vs. uninfected hepatocytes. Finally, accumulated HBx transactivates transcription of cellular genes involved in the regulation of cell proliferation, such as c-jun, c-fos, and c-myc, etc., and the transcriptional transactivation appears to be accompanied by the stimulation of the protein kinase C and NF-κB pathways [138]. These HBV-associated disruptions of signaling cascades consequently foster progression of HCC and are thoroughly reviewed in Levrero et al. [139]. 

As described above, HCV differs significantly in its structure, life cycle, and epidemiology. HCV infection is heavily distributed in some African and Asian countries, not in North America and Western Europe, and the number of HCV-infected people is lower than that of HBV-infected hosts [140,141]. Nonetheless, numerous previous reports indicated that HCV shares molecular processes and pathogenesis with HBV in many aspects leading to hepatocarcinogenesis [140,142]. That is, both HBV and HCV promote inflammatory reactions and oxidative stress. Liver injury by these agents sequentially leads to fibrosis, cirrhosis, and HCC [143]. Since the genotype and replicability of HBV relate to the risk of developing HCC [58,80], a meta-analysis of large cohort studies in both cirrhotic and non-cirrhotic patients also indicates that genotype 1b of HCV carries a 2.5-fold higher risk of HCC than patients with other genotypes [144,145,146,147], indicating strain-specific differences in oncogenic potential. Further, as observed in HBV-associated HCC, HCV infection of liver cells generates extensive heterogeneity of genomic alterations. For example, approximately 15–50% of p53 and 20–40% of β-catenin is mutated in HCV-associated HCC [148]. β-catenin plays a critical role in the regulation and coordination of cell–cell adhesion [149,150,151], gene transcription [151], and tumorigenesis [152,153,154]. Like HBx, the HCV core protein is known to cause genomic instability in viral and cellular genes, which is critical for the transformation of the murine fibroblasts and tumorigenesis [155]. Expression of the core gene is closely associated with the immortalization of the primary hepatocytes, leading to cellular transformation and carcinogenesis [155,156,157,158]. Sequence analysis confirmed the gene mutations increased in HCC [159,160,161,162]. Other viral genes mutations, such as NS5A, have also been associated with the risk of HCC [160,163].

Signaling cascade dysregulation. These mutations in the viral and cellular genes of HCV-infected cells affect various signaling cascades that play an important role in the progression of HCC. Chronic infection with HCV determines the fate of infected cells by regulating signaling cascades involved in cell proliferation and death. As was observed during HBV infection, the HCV core protein activates the Ras/Raf/mitogen-activated protein kinase/extracellular-signal-regulated kinase pathway, whose signaling cascades are essential for the regulation of apoptosis or cell cycle progression [164,165]. The HCV core protein augments cytoplasmic p21, while suppressing nucleic p21 by the p53-dependent pathway, to determine the fate of the infected cells with respect to the proliferation and apoptosis [166,167,168]. Meanwhile, NS5A downregulates growth arrest and DNA-damage-inducible gene 45-a (GADD45a) through the NF-κB and phosphatidylinositol-3-kinase (PI3K)/Akt pathways to regulate cell proliferation [169]. HCV NS5A is also known to block caspase-dependent and mammalian target of rapamycin (mTOR)-mediated apoptosis [161,170], indicating that the NS5A-mediated PI3K/Akt-mTOR pathway plays a pivotal role in the control of cell survival. The WNT/β-catenin pathway is also regulated by HCV infection, just as the pathway is modulated with HBV and alcoholic liver cirrhosis [148]. HCV NS5B, like HBx and pre-S2 mutation in HBV, inhibits the tumor suppressor Rb signaling cascade, which in turn promotes the progression of the cell to the next phase of the cell cycle. The HCV core protein also regulates cell cycles by activating various G1 phase proteins, such as CDK2, and inhibits p21Cip1 (a CDK inhibitor) [171,172]. Signaling transductions leading to increases in oxidative stress by chronic HCV infection are also affected by increased oxidative stress markers, such as lipid peroxidation products, superoxide dismutase, etc. [173]. HCV NS5A disturbs calcium signaling and increases mitochondrial ROS, leading to translocation of NF-κB and STAT3 to the nucleus [174], which is required for cytokine signaling and its activation. HCV viral proteins also regulate the expression of the cellular proteins important for the HCV life cycle. HCV protease NS3/4A enhances transcription of epidermal growth factor receptor, which plays a key role in HCV entry processes by redistributing claudin-1 [175,176,177], and reduces the intrahepatic production of interferon (IFN) γ (IFNγ) and TNFα, impairing the hepatic inflammatory response against HCV infection [178,179]. Since both HBV and HCV share common features crucial for hepatocarcinogenesis, both viruses cause HCC, even as they display distinct life cycles in the infected hepatocytes.

#### 2.2.2. HIV-1/HBV or /HCV Co-Infection

In co-infected patients, every stage of HCV-mediated liver disease progression is accelerated [180,181,182,183]. Thus, co-infection in Western countries has become a leading cause of morbidity and mortality in HIV-1-infected individuals [184,185,186]. However, the molecular mechanisms accelerating liver disease during co-infection are poorly studied. One possible mechanism may occur through direct HIV-1 infection of HBV- and HCV-infected hepatocytes followed by stimulation of HBV and HCV replication, resulting in increased pro-inflammatory responses [187,188,189]. Replication of HIV-1 in hepatocytes is controversial [190,191,192,193,194,195], as the expression of receptor and co-receptor molecules necessary for HIV-1 entry into hepatocytes is disputed [42,43,44,196]. Alternatively, HIV-1 viral proteins and/or cellular elements may indirectly trigger HBV- and HCV-infected hepatocytes to enhance replication. Another possibility is that activation of immune cells and dysregulation of expression and secretion of pro-inflammatory cytokines can induce liver problems, such as steatosis [197,198,199,200], fibrosis [198], etc. However, it is currently unclear which elements are the most important mediators of these actions and how HIV-1 and/or viral proteins induce alterations to promote hepatocarcinogenesis. Below we will discuss the potential interplay of different HIV-1 proteins as mediators of HCC pathogenesis among HIV-1/HBV or /HCV co-infected patients (Table 2) and possible influences of HIV antiviral therapies.

Since HIV-1 Tat protein is diffusible [39], the protein secreted from HIV-1-infected cells could diffuse into hepatocytes to enhance HBV/HCV replication and expression of hepato-cellular genes, thus promoting liver disease. Indeed, HIV-1 Tat is known to enhance hepatocarcinogenesis in transgenic mice [40,41]. The HIV-1 envelope gp120 could directly interact with hepatocyte CXCR4 or CCR5 co-receptor molecules [42,43] and trigger signals that modulate the expression of cellular and/or HBC/HCV genes. However, these data require further evaluation, since recent reports indicate that hepatic cell CXCR4, CCR5, and CD4 expression is limited [44,45], which weakens the case for Env as a critical mediator of HIV-1/HBV or /HCV co-infection pathogenesis. HIV-1 Nef can be transferred from HIV-1-infected cells to non-susceptible cells, including hepatocytes, through conduits (filopodia) and/or exosomes [46,47,48,49]. Nef could also modulate HCV replication by regulating the amount of intracellular lipid molecules that are essential for efficient HCV replication at the replication site [201,202,203,204,205,206]. Further, Nef forms complexes with and thereby activates several cellular kinases, such as the Src family of tyrosine kinases [207], through the proline-rich domain of Nef [48,207], which could promote tumorigenesis. Finally, liver disease is enhanced by immunologic alterations [199,200,208] during HIV/HBV or /HCV co-infection, and Nef is known to contribute to many of these immunologic alterations [198,208,209,210,211,212,213]. These data strongly indicate that HIV-1 Nef could be a critical viral factor exacerbating liver disease in people living with HIV-1 and hepatitis. Nonetheless, to date, there has not been comprehensive research in the area of Nef-mediated pathogenesis of hepatic disease.

Therapeutic interventions for HIV-1/HBV co-infections rely on reverse transcriptases with structurally similar enzymatic pockets and thus are susceptible to the same enzymatic inhibitors [214]. The Food and Drug Administration approved four nucleoside reverse transcriptase inhibitors (NRTI) that target HIV, i.e., lamivudine, emtricitabine, tenofovir disoproxil fumarate, and tenofovir alafenamide, for use in HBV. While these NRTI impede HBV polymerase activity and viral replication [215], they demand potentially life-long regimens and are not curative. Accordingly, clevudine-triphosphate, which shares functionalities with both NRTI and non-nucleoside reverse transcriptase inhibitors, has been introduced to address the weaknesses of NRTI [216]. In HIV-1/HCV co-infections, HIV-targeted cART is ineffective at reducing HCV replication. Instead, pan-genotypic direct-acting antivirals (i.e., NS5A inhibitors and NS3/4A protease inhibitors) are used to cure most persons of HCV infection in 12–24 weeks depending on the state of liver fibrosis [217]. However, ART and HCV treatment regimens must be selected with special consideration for potential drug–drug interactions and overlapping toxicities as many combinations are counter-indicated [218]. Direct-acting antivirals are a dramatic improvement over previous interferon-based therapies that had significant adverse effects and counterindications for PLWH [217,219,220]. In both HIV-1/HBV and HCV co-infections, the benefits of cART outweigh the risks of co-therapies. However, the need for anti-co-infection therapeutics capable of targeting oncogenic viruses such as HBV/HCV without detrimental effects on HIV suppression or drug-induced toxicity remains the highest priority.

### 2.3. Human Papillomavirus (HPV)

Transmission and entry. As a member of the Papovaviridae family, HPV is a small, non-enveloped double-stranded DNA virus [221,222] that is primarily transmitted by skin-to-skin or skin-to-mucosa contact. Sexual contact, such as vaginal or anal sex with someone living with HPV, is the most common type of transmission and the virus can be transferred without penetrative sex [223,224]. To gain entry HPV binds to the basement membrane and then to the cell surface through the interactions between the viral capsid protein L1 and heparin sulfate proteoglycans (HSPGs) on the surface of the target cells, which are principally the squamous epithelium of the human mucosa and skin [225,226,227,228,229]. The receptor-bound virus will then be internalized into the cell via clathrin-mediated endocytosis [230,231]. Notably, other studies indicate that internalization of HPV can be dependent [232] or independent [233] of clathrin- and caveolin-mediated endocytosis.

Life cycle. Once internalized HPV migrates from the early endosome to the nucleus with the help of the L2 protein. Host cellular components play an essential role in intracellular viral trafficking as HPV passes from late endosomes/lysosomes to the trans-Golgi-network, Golgi apparatus, and the ER prior to establishing itself in the nucleus as low copy, viral DNA episomes [234]. The episomal HPV genomes are then quickly replicated to generate HPV early proteins (Figure 5), including E1 and E2 as well as E6 and E7, from the early viral promoter, marking an initial amplification phase [235]. These early proteins are essential for the overall regulation of the HPV life cycle, including genome replication, gene expression, evasion of the host immune systems, and viral genome persistence [236]. The E1 and E2 proteins function as replication factors by binding to the replication origin sequences located near the start of early transcription. Complex formation of E1 and E2 with the replication origin facilitates recruitment of cellular replication factors to the viral origin [237,238]. E2C, a truncated form of E2, acts as a viral transcription repressor to modulate expression levels throughout the HPV life cycle [239]. E4 contributes to genome amplification efficiency and virus synthesis as well as expedites virion egress following assembly in suprabasal cells [240]. E5 regulates the HPV-infected cell life cycle by delaying normal epithelial cell differentiation and maintaining cell cycle progression [241]. Functions of E6 and E7, the two major viral transforming proteins, are described in the following section.

Replication of HPV genomes in synchrony with chromosomal DNA replication in the S phase of infected basal cells generates new genomes that will be equally distributed to two daughter cells. One of the two daughter cells moves toward suprabasal layers and undergoes differentiation. Differentiation switches replication of the viral genome to a productive mode, dramatically increasing expression of E1 and E2, and thus genomic copies [235,242]. The structural major and minor capsid proteins, L1 and L2, respectively (Figure 5), will then be expressed by the late promoter in the terminally differentiated layer of the epithelium [235,242]. The L1/L2 capsid will then be assembled to generate a mature virus particle, which is shed together with the dead squamous cells of the exterior-most epithelium for the next transmission journey [235].

#### 2.3.1. Molecular Mechanisms on HPV-Mediated Carcinogenesis

Over 400 different types of HPV have been identified. However, only a small subset of HPV strains is characterized by persistent infection that generates squamous proliferative lesions, also called verruca/wart, papilloma, and condyloma [243,244]. Among them, types 16 and 18 belong to the α-papilloma virus and are the main causes of HPV-associated cancers. Approximately 50% of cervical cancer cases are associated with HPV16, 25% of cases are linked with HPV18, and the remaining 25% of cases are caused by the remaining HPV types [245]. Activation of E6 and E7 induces degradation of tumor suppressor molecules, p53 and Rb, respectively, to accelerate the progression of HPV-mediated tumorigenesis. Briefly, E6 interacting with E6-associating protein (E6-AP, also known as UBE3A), an E3 ubiquitin ligase, functions as a carrier protein to recruit p53 to the proximal position of E6 and E6-AP complex so that E6-AP degrades p53 via the ubiquitin proteosome system [246,247]. Likewise, the binding of E7 to Rb releases the E2F transcription factor from Rb/E2F complex to promote cell cycle progression and induces E7-mediated proteasomal degradation of Rb [248,249]. Calpain, another protease, also plays an important role in the rapid degradation of Rb by cleaving Rb to expose a region better recognized by the ubiquitin proteosome system [250]. E6 and E7 also increase transforming activity by binding several other cellular factors, such as PDZ proteins and p600 [251,252]. Further, these oncoproteins suppress the host immune responses by blocking the IFN pathways and thereby enabling the virus to evade the host immune surveillances. In brief, E6 blocks the Jak-STAT pathway initiated by IFNα by physical association with tyrosine kinase 2 [253], while E7 inhibits the antiviral and anti-growth functions of IFNα and inactivates IFN regulatory factor-1, a tumor suppressor protein. All of which enhance the persistent infection of high-risk HPVs to initiate neoplastic change.

The integration of HPV into the human genome is a key genetic event in cervical carcinogenesis. Not only does integration generate insertional mutagenesis, but it also induces persistent expression of the viral oncoproteins, E6 and E7 [254], to augment the progression of HPV-mediated tumorigenesis. It is reported that HPV type 16 integration in cervical cancers frequently takes place in common fragile sites [255]. A recent whole-genome sequencing and high-throughput viral integration detection analysis identified the clustered genomic hot spots that promote the integration of HPV in cervical cancer [256]. Further, the study also indicated that HPV integration into the host genome may have occurred by microhomology-mediated DNA repair pathways. Moreover, the integration altered the expression of several key proteins, such as fragile histidine triad (FHIT), Myc, high mobility group AT-hook 2 (HMGA2), etc., which play an important role in tumorigenesis [256,257]. Taken together, these reports demonstrate that HPV also causes cancers by insertional mutagenesis, which dysregulates the expression of viral and cellular genes playing a pivotal role in the tumorigenesis.

#### 2.3.2. HIV-1/HPV Co-Infection

There is a statistically significant excess of HPV-associated cancers in AIDS patients compared with those in non-AIDS patients [53]. Specifically, HIV-1 infection in HPV-infected patients imparts a 2–4-fold increase in risk for head and neck squamous cell carcinoma, a 2-fold increase for anal cancer, [53,54], and a 3-fold increase for cervical cancer [8]. The intersection of cART on HPV is difficult to define, perhaps because of the heterogeneity and ubiquity of HPV infection in the world population. A recent study demonstrated that amprenavir, a protease inhibitor discontinued in 2007, compromised cell–cell junctions in a stratified model of the oral or cervical epithelium. In this model, protease inhibitors increased the permeability of HPV16 into the basal layer to promote infection and viral spread [258]. However, the effect of amprenavir on HPV progression in PLWH has not been examined. Many studies have not found an association between cART, HIV, and HPV incidence, prevalence, or progression [259,260,261]. However, an HIV Epidemiology Research Study found an increased HPV clearance rate in HIV-positive women diagnosed with squamous intraepithelial lesions on cART compared to HIV-negative women [262]. Further, a meta-analysis of cross-sectional, cohort studies of women living with HIV between 1996 and 2017 revealed that early cART initiation and sustained adherence reduces the incidence and progression of HPV-associated lesions [263]. Regardless, there is no cure for HPV, and the interplay between HIV-1 infection, cART, and HPV progression remains unclear. Thus, to effectively suppress dual infections, it is imperative that pharmaceuticals that simultaneously target both HIV-1 and HPV are developed or that cART must be safely co-administered with anti-oncovirus therapies.

However, the molecular mechanisms by which HIV-1 exacerbates HPV-mediated cancers in HIV/HPV co-infected patients are currently unknown. Human epithelial cells, and more specifically, basal keratinocytes, are exclusive targets for HPV [264,265], while CD4+ immune cells, such as T helper cells, monocyte/macrophages, dendritic cells, etc., are the major targets for HIV-1 infection [266,267]. While previous reports indicated that HIV-1 can traverse the epithelial lining [268,269] or that HIV-1 antigens were detected in the epithelial cells [270], there have not been adequately convincing data for HIV-1 infection of HPV-target cells. Since HIV-1 and HPV are unlikely to encounter one another in infected cells, co-infection as a mechanism to aggravate HPV-associated cancers in co-infected patients is also not feasible. How then does HIV-1 exacerbate HPV-induced cancers in people living with HIV-1 and HPV comorbidities?

Rather than direct interaction of HIV-1 and HPV infection within individual target cells to expedite the progression of HPV-mediated cancers, transferrable HIV-1 viral proteins from HIV-1-infected cells to HPV-infected cells could be integral to exacerbating HPV-associated tumorigenesis, similar to that discussed in HCC (Table 3). HIV-1 Tat augments the expression of HPV oncoproteins E6 and E7, increasing E6- and E7-mediated oncogenic effects HIV-1 Tat also increases HPV virion protein L1, which mediates initial attachment of the virion to target cells, raising the cellular infectivity of HPV [271,272]. Expression of HPV L1 is also enhanced by HIV-1 Rev [273], even though it remains unknown how Rev reaches HPV-infected cells given its nuclear localization in HIV-1-infected cells. Viral protein R (Vpr), another nuclear protein involved in cell cycle regulation [274,275,276], interacts with E6 in cervical cancer cells [277]. However, again, it is unclear how and where Vpr encounters E6 for cell cycle regulation. HIV-1 Nef, a known transferable protein from HIV-1-infected cells to the neighboring target cells via filopodia and/or exosomes [46,47,48,49], interacts with and degrades E6-AP (HPV oncoprotein E6-associating protein) [278]. This interaction is crucial for the degradation of tumor suppressor, p53, via interacting with E6 in HPV-infected cells. Other reports also indicated that HIV-1 Nef degrades p53 [279,280], suggesting that Nef could be a potential regulator of p53 stability. Dysregulation of miRNA expression may also contribute to the pathogenesis of anal cancer [281]. Moreover, HIV-1 infection-associated dysregulation of host immune responses may directly or indirectly protect HPV-infected keratinocytes from cytotoxic T-lymphocyte-mediated killing [281].

## 3. Discussion

Therapeutic intervention for people living with HIV-1 with cART can reduce a patient’s viral load so dramatically that the virus becomes undetectable in the blood. Decreased viral load, in turn, lessons damage to the immune system, such as CD4 count, which is depleted by HIV-1 infection. Thus, the advent of cART has significantly reduced the risks of oncovirus-related AIDS-defining cancers in people living with HIV-1; however, those of non-AIDS-defining cancers have moved in the opposite direction. For instance, as noted above, patients with co-infection of HIV-1/HCV are at higher risk for HCC than those with HCV mono-infection [282]. However, it remains uncertain how cART treatment exerts differential consequences on AIDS-defining vs non-AIDS-defining cancers in people living with co-infections. Further, the molecular processes by which HIV-1 exacerbates non-AIDS-defining cancers require additional investigation.

Dramatic amelioration of AIDS-defining cancers but not non-AIDS-defining cancers by cART treatment indicates that immunologic status is not closely linked to non-AIDS-defining cancers. However, it is unknown how cART, which suppresses HIV-1 replication followed by improvement of host immune responses, then brings about differential outcomes in raising those cancer risks. It is postulated that cART toxicity could play a role in the differential determination of the cancer risk in people living with HIV [283,284,285,286]. Specifically, the cART toxicity exposure period is related to the cancer risks, i.e., the risks of AIDS-defining cancers, such as Kaposi sarcoma and non-Hodgkin lymphoma, were clearly reduced by cART treatment during early HIV-1 infection, before the development of overt immunosuppression [287,288]. However, the impact of long-term cART exposure on the risk for non-AIDS-defining cancers has not been well defined, albeit some studies have demonstrated long-term exposure of cART to be independently associated with non-AIDS-defining malignancies in the HIV-1 clinical setting [283,284,285,289]. 

Inefficacies of antiretrovirals-only combination therapy for reducing morbidity and mortality in non-AIDS-defining cancers now argue for either the development of pharmaceuticals that each simultaneously target the two co-infecting viruses or the administration of antiretrovirals combined with anti-oncovirus drugs to counter these dual infections. It will similarly be of great interest to discover anti-co-infection therapeutics that target various other oncoviruses without eliciting appreciable drug resistance or side effects, in the quest for safely and effectively treating HIV-1 and non-AIDS-defining cancers. The elucidation of the relationship between cART exposure and non-AIDS-defining cancers incidence, and of cART-associated molecular mechanisms differentially modulating AIDS- and non-AIDS-defining cancer risks, will be essential for reducing the morbidity and mortality of non-AIDS-defining cancers in co-infected patients.

It is proposed that HIV-1 co-infection increases the risks of oncovirus-mediated tumorigenesis in several different ways (Figure 1). First, HIV-1-infection of oncovirus-infected target cells could directly enhance oncogenic effects. However, as reviewed above, HIV-1 and oncoviruses employ disparate receptor/co-receptor molecules for entry into host cells (Table 1), and thus their target cells are distinctively different. Nonetheless, numerous reports assert that HIV-1can infect oncovirus target cells, such as hepatocytes and epithelial cells, to exacerbate the oncovirus-mediated tumorigenesis. Alternatively, transferrable HIV-1 proteins, such as gp120, Tat, Nef, and Vpr, to the neighboring uninfected cells could be responsible for expediting the progression of co-infection-associated cancers [272,290].

Immunosuppression could be another important compounding factor accelerating the progression of tumors. There is a tight link between immune suppression and the risk of cancers. Recovery of immune responses, such as increased CD4 T cell counts with effective cART, clearly lowers the risk of AIDS-defining cancers such as Kaposi sarcoma and non-Hodgkin lymphoma. However, the relationship between HIV-associated immunosuppression and the risk of non-AIDS-defining cancers, including HCC, cervical cancer, and anal cancer is less clear [291,292,293]. Increased risk for non-AIDS-defining cancers despite widespread cART use could result from an incomplete restoration of the immune system. Nadir CD4 levels during HIV-1 associated viremia and subsequent cART-mediated immune reconstitution could foster permanent immune alterations, which are involved in malignancy and cannot be determined by CD4 counts alone [294].

Finally, HIV-1 infection induces various types of inflammatory response, and the heightened inflammation is associated with malignancy. Progression of AIDS and rapid drop of CD4 count augment levels of inflammation. The high level of inflammation can be attenuated but not normalized by cART treatment [295,296]. A small but statistically significant association of IL-6 and C-reactive protein is linked to developing infection-related and -unrelated malignancies. However, it is unclear how HIV-induced inflammatory reactions exert differential impacts on the progression of AIDS-defining and non-AIDS-defining cancers.

In view of the unsolved molecular pathology, it is imperative to elucidate the molecular mechanisms for how HIV-1 infection in the oncovirus-infected hosts accelerates the oncogenic virus-mediated tumors, which is paramount for developing therapeutics and deterrents against cancer progression. Further exploration of the impacts of antiviral therapies and vaccination on chronic infection and tumorigenesis is required to fully understand non-AIDS-defining cancer in people living with HIV and other oncogenic viruses.

## Figures and Tables

**Figure 1 biomedicines-10-00768-f001:**
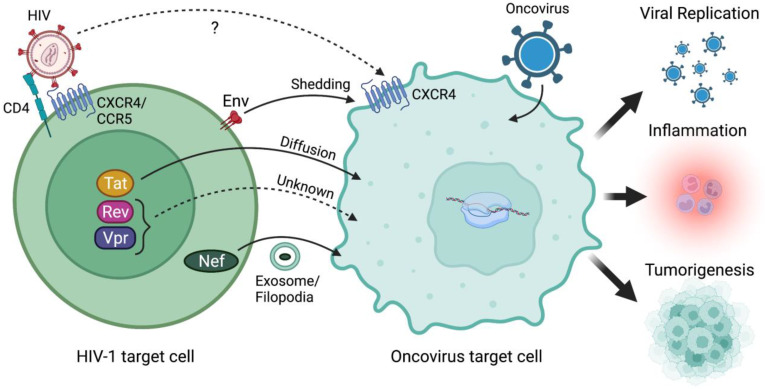
HIV-1 and/or its viral proteins accelerate oncovirus-mediated tumorigenesis. It remains controversial whether HIV-1 can infect the same target cells as oncoviruses given the absence of essential receptor/co-receptor expression needed for HIV-1 binding and entry into oncovirus target cells. Alternatively, HIV-1 proteins which can be transferred from an HIV-1-infected cell to an oncovirus target cell via mechanisms such as shedding (Env), diffusion (Tat), or exosome/conduit-mediated assistance (Nef) could instead be accelerating oncovirus-mediated tumorigenesis. Notably, the cell–cell transfer mechanisms of HIV-1 Rev and Vpr proteins remain unknown. Regardless, HIV-1 infection and/or transfer of HIV-1 proteins to oncovirus-infected target cells can promote increased oncovirus replication and infectivity, which leads to increased pathogenicity propagated by the oncovirus. Increased viral replication, co-infections, and HIV-1 proteins can alter inflammatory and immune responses in tumor microenvironments which can promote survival signals and even tumor growth. Finally, HIV-1 proteins themselves can modulate gene expression to upregulate oncogenic factors as well as interact with cell signaling pathways that regulate cell fate including cell cycle, growth/proliferation, migration, inflammation, and survival to enhance tumorigenesis. Created with BioRender.com (accessed on 28 January 2022).

**Figure 2 biomedicines-10-00768-f002:**
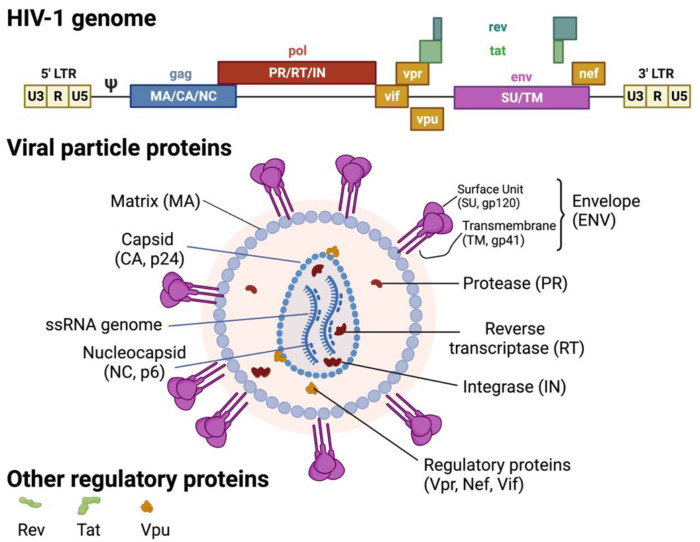
HIV-1 genome and virion structure. Created with BioRender.com (accessed on 28 January 2022).

**Figure 3 biomedicines-10-00768-f003:**
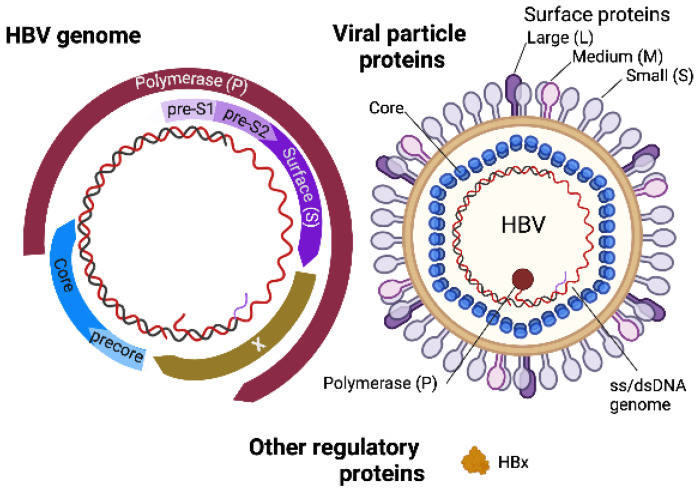
HBV genome and virion structure. Created with BioRender.com (accessed on 28 January 2022).

**Figure 4 biomedicines-10-00768-f004:**
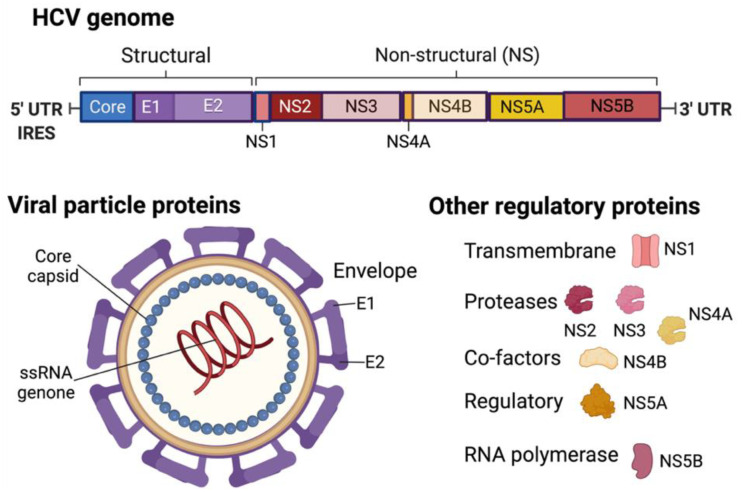
HCV genome and virion structure. Created with BioRender.com (accessed on 28 January 2022).

**Figure 5 biomedicines-10-00768-f005:**
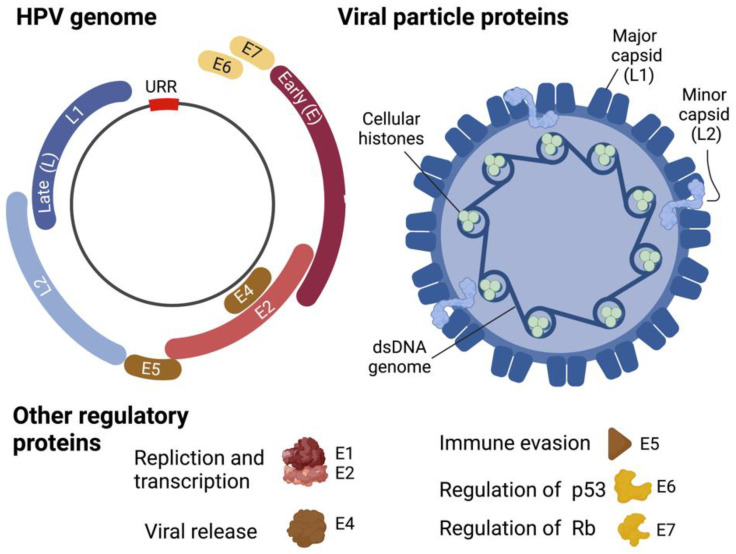
HPV genome and virion structure. Created with BioRender.com (accessed on 28 January 2022).

**Table 1 biomedicines-10-00768-t001:** Summary of HIV-1 and oncoviruses inducing non-AIDS-defining cancers.

Virus	Family	Viral Genome	Receptor/Co-Receptor	Primary Target Cells	Tumor Types
HIV-1	Retroviridae	ssRNA	CD4/CXCR4 CD4/CCR5	CD4+ T cells, macrophages, monocytes, dendritic cells	
HBV	Hepadnaviridae	ss/dsDNA	NTCP	Hepatocytes	Hepatocellular carcinoma
HCV	Flaviviridae	ssRNA	CD81, SR-B1, CLDN1, OCLN, etc.	Hepatocytes	Hepatocellular carcinoma
HPV	Papillomaviridae	dsDNA	HSPG	Epithelial cells	Cervical, oropharyngeal, anal, genital cancers, etc.

Abbreviations: Cluster of differentiation 4 (CD4)/C-X-C motif chemokine receptor 4 (CXCR4), C-C motif chemokine receptor 5 (CCR5), sodium taurocholate co-transporting polypeptide (NTCP), CD81, Scavenger receptor class B type I (SR-BI), claudin-1 (CLDN1), and occludin (OCLN), Heparin sulfate proteoglycans (HSPGs).

**Table 2 biomedicines-10-00768-t002:** HIV-1 and HBV/HCV protein interactions.

Oncovirus	HIV-1 Protein	Mechanism	Outcome	Refs.
HBV/HCV	Tat	Unknown	Modulate oncogene expression	[40,41]
	Env	Co-receptor interaction	Dysregulate co-receptor mediated signaling cascades	[42,43]
	Nef	Lipid modulation	Enhance viral replication	[201,202,203,204,205,206]
		Activates cellular kinases	Dysregulate growth and survival signaling cascades	[48,207]
		ROS and other unknown	Immunologic alterations	[198,208,209,210,211,212,213]

**Table 3 biomedicines-10-00768-t003:** HIV-1 and HPV protein interactions.

Oncovirus	HIV-1 Protein	Mechanism	Outcome	Refs.
HPV	Tat	Increases expression of oncoproteins (E6, E7)	Increases E6- and E7-mediated oncogenic effects	[271,272]
		Increases expression of virion protein (L1)	Enhance infectivity of HPV	[271,272]
	Rev	Increases expression of virion protein (L1)	Enhance infectivity of HPV	[273]
	Vpr	Interacts with E6	Enhance infectivity of HPV	[277]
	Nef	Interacts with and degrades E6-AP	Regulates cell cycle arrest	[278]
		Degradation of tumor suppressor, p53	Promotes p53 ubiquitination & degradation	[279,280]
	Unknown	Dysregulation of miRNA expression	Regulate oncogenic expression	[281]

## Data Availability

Not applicable.

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
