# Peer review of "HIV-1-Mediated Acceleration of Oncovirus-Related Non-AIDS-Defining Cancers"

_biomedicines, 2022, doi:10.3390/biomedicines10040768_

Round 1

Reviewer 1 Report

The manuscript entitled “HIV-1-mediated acceleration of oncovirus-related non-AIDS- 2 defining cancers” by Proulx et al have described/reviewed about the genetics and life cycle of HIV, and other oncoviruses (HBV, HCV, and HPV), molecular mechanism of individual oncoviruses, and mechanism of each oncoviruses with HIV-1 co-infection. The manuscript is well written with good number of literatures. However, the manuscript is not well structured for the objective. My major concern is as follows

Description of overview of HIV-1 and other oncovirus looks very fundamental and most of the researchers knew the genetics and life cycle of those viruses. If Authors feel overview of those virus is required, please cut short to single paragraph for each virus. In addition, Figures for genome and virion structure for those viruses are not relevant to the objective of this review except figure 1. Instead, Authors should focus on figures that clearly depicts the mechanism of individual oncoviruses and with HIV-1 co-infection that contribute to cancer disease or figures depicting the modification/alteration of cell pathways by the oncoviruses/HIV leading to cancer.

Present the interaction of genes/proteins of HIV with proteins of individual oncoviruses that accelerates for non-AIDS defining cancer as a table/figure.

We have sufficient evidence for other oncoviruses such as EBV and HHV that are very common for non-AIDS defining cancers in patients with HIV and is not discussed.

Minor comments

Giving brief subtitles for the different mechanisms instead of first, second, so on under “Molecular mechanisms on HBV- and HCV-mediated hepatocarcinogenesis” ease the understanding of readers.

Line 127-128: Remove the sentence “approximately 5 % of adults with chronic HCV infection are also co-infected with HIV-1” as this observation is made from the study conducted only in 15 states of USA. In addition, the sentence should be other way around.

Line 128-129 and 327-329: Sentence duplication. In addition, quoting references for the same sentence differs. Merge the reference and keep the sentence either one of the places.

Line 486-487: Did Authors mean “Thus, the advent of cART has significantly reduced the risks of oncovirus-related AIDS defining cancers in people living with HIV-1”?

Author Response

Reviewer #1:

  • Description of overview of HIV-1 and other oncovirus looks very fundamental and most of the researchers knew the genetics and life cycle of those viruses. If Authors feel overview of those virus is required, please cut short to single paragraph for each virus. In addition, Figures for genome and virion structure for those viruses are not relevant to the objective of this review except figure 1. Instead, Authors should focus on figures that clearly depicts the mechanism of individual oncoviruses and with HIV-1 co-infection that contribute to cancer disease or figures depicting the modification/alteration of cell pathways by the oncoviruses/HIV leading to cancer.

[Response] We feel inclusion of virus overviews and individual figures enhances readership and clarity to all audiences. We agree additional depiction of HIV and oncovirus interactions will enhance manuscript. We have now added two tables that illustrate HIV-1 protein interactions in (1) HBV/HCV tumorigenesis and (2) HPV tumorigenesis.

  • Present the interaction of genes/proteins of HIV with proteins of individual oncoviruses that accelerates for non-AIDS defining cancer as a table/figure.

[Response] We agree additional depiction of HIV and oncovirus interactions will enhance manuscript. We have now added two tables that illustrate HIV-1 protein interactions in (1) HBV/HCV tumorigenesis and (2) HPV tumorigenesis.

  • We have sufficient evidence for other oncoviruses such as EBV and HHV that are very common for non-AIDS defining cancers in patients with HIV and is not discussed.

[Response] Thank you for this point. We agree discussion of EBV and HHV are warranted. We have added the following paragraph to the end of section 2.2:

There are conflicting reports on whether EBV causes AIDS-defining or non-AIDS-defining cancers.  It is reported that EBV causes lymphoproliferative disease, including AIDS-defining lymphomas such as Burkitt’s lymphoma as well as HIV-1 related Hodgkin lymphoma (HL) (Front. Oncol. 10:1723), while other reports indicate that EBV induces non-AIDS-defining cancer, such as HL (AIDS Res Ther (2018) 15:16; World J Virol. 2015;4:209).  HHV also induces AIDS-defining, not non-AIDS-defining cancer. 

We have added the following paragraph to the end of section 2.2: “This review is focused only on oncogenic virus-mediated non-AIDS-defining cancers, excluding AIDS-defining cancers, such as EBV and HHV.”

  • Giving brief subtitles for the different mechanisms instead of first, second, so on under “Molecular mechanisms on HBV- and HCV-mediated hepatocarcinogenesis” ease the understanding of readers

[Response] Brief subtitles were added to ease the understanding of readers. We appreciate the suggestion.

  • Line 127-128: Remove the sentence “approximately 5 % of adults with chronic HCV infection are also co-infected with HIV-1” as this observation is made from the study conducted only in 15 states of USA. In addition, the sentence should be other way around.

[Response] We have removed this statement.

  • Line 128-129 and 327-329: Sentence duplication. In addition, quoting references for the same sentence differs. Merge the reference and keep the sentence either one of the places.

[Response] We have removed the latter sentence duplication and merged the references. 

  • Line 486-487: Did Authors mean “Thus, the advent of cART has significantly reduced the risks of oncovirus-relatedAIDS defining cancers in people living with HIV-1”?

[Response] Antiretroviral therapy has reduced the incidence of oncovirus-related AIDS defining cancers.  As discussed above those related to EBV and HHV are significantly reduced in the post-ART era.    

Reviewer 2 Report

The review aims to investigate the HIV mediated oncovirus cancers and provides a brief summary of HIV and come cancer-causing viruses. 

Given that the abstract started with a discussion on ART, it causes the reader to expect sections and in-depth discussions to how the ARTs for specific viruses, specifically HIV will impact the other oncoviruses.

The figures generated from Biorender is nice, and it would be good to have an overarching figures to illustrate the interaction of HIV and the other mentioned viruses.

This is in fact the lacking of the review - the molecular mechanisms of HIV and the oncoviruses, whether mediated by ART or not, should be emphasized. At this moment, the review just reads of disparate sections of the viruses, with some mention of coinfection, but without in-depth discussion. Coinfections are not exciting, just as there are HIV-infected patients facing superinfections of COVID19.

The authors are strongly advised to discuss in-depth the immune/ART cross-interaction (or lack thereof) of HIV with these oncoviruses and provide novel insights and discussion beyond the very obvious of CD4 Tcell count decrease with HIV infection.

Minor:

Some areas of spacing e.g. line 348 467

co-infection or coinfection - need to be consistent.

Author Response

  • Given that the abstract started with a discussion on ART, it causes the reader to expect sections and in-depth discussions to how the ARTs for specific viruses, specifically HIV will impact the other oncoviruses… The authors are strongly advised to discuss in-depth the immune/ART cross-interaction (or lack thereof) of HIV with these oncoviruses and provide novel insights and discussion beyond the very obvious of CD4 Tcell count decrease with HIV infection.

[Response] We fully accept the reviewer’s point and hence thoroughly discuss the potential impact of cART on NADC by adding the following paragraph to the end of the first paragraph of the Discussion:

"Dramatic amelioration of AIDS-defining cancers but not non-AIDS-defining cancers by cART treatment indicates that immunologic status is not closely linked to non-AIDS-defining cancers.  However, it is unknown how cART, which suppresses HIV-1 replication followed by improvement of host immune responses, then brings about differential outcomes in raising those cancer risks.  It is postulated that cART toxicity could play a role in the differential determination of the cancer risk in people living with HIV [270-273]. Specifically, the cART toxicity exposure period is related to the cancer risks, i.e. the risks of AIDS-defining cancers, such as Kaposi sarcoma and non-Hodgkin lymphoma, were clearly reduced by cART treatment during early HIV-1 infection, before the development of overt immunosuppression [274,275].  However, the impact of long-term cART exposure on the risk for non-AIDS-defining cancers has not been well defined, albeit some studies have demonstrated long-term exposure of cART to be independently associated with non-AIDS-defining malignancies in the HIV-1 clinical setting [270-272,276].  Thus, elucidation of the relationship between cART exposure and non-AIDS-defining cancers incidence, and of cART-associated molecular mechanisms differentially modulating AIDS- and non-AIDS-defining cancer risks, will be essential for reducing the morbidity and mortality of non-AIDS-defining cancers in co-infected patients."

  • The figures generated from Biorender is nice, and it would be good to have an overarching figures to illustrate the interaction of HIV and the other mentioned viruses. This is in fact the lacking of the review - the molecular mechanisms of HIV and the oncoviruses, whether mediated by ART or not, should be emphasized. At this moment, the review just reads of disparate sections of the viruses, with some mention of coinfection, but without in-depth discussion. Coinfections are not exciting, just as there are HIV-infected patients facing superinfections of COVID19.

[Response] We agree additional depiction of HIV and oncovirus interactions will enhance manuscript. We have now added two tables that illustrate HIV-1 protein interactions in (1) HBV/HCV tumorigenesis and (2) HPV tumorigenesis.

  • Some areas of spacing e.g. line 348 467

[Response] Thank you. Spacing is resolved.

  • co-infection or coinfection - need to be consistent.

[Response] All mentioning of ‘coinfection’ in text have been changed to ‘co-infection’

Round 2

Reviewer 1 Report

References cited in the abbreviation (Table 1) is not required and should be removed.

What is meant by unchecked tumor growth and survival in table3? Provide the appropriate outcome for the interaction of HIV-1 protein and HPV protein. For example, “HIV-1 Vpr induces cell-cycle arrest with expression of HPV-16 E6”

In table 2 and 3: Providing the unknown mechanism with unknown HIV-1 protein does not make any sense.

Author Response

Reviewer #1:

  1. References cited in the abbreviation (Table 1) is not required and should be removed.

Response: Thank you for the suggestion, these references have been removed.

  1. What is meant by unchecked tumor growth and survival in table3? Provide the appropriate outcome for the interaction of HIV-1 protein and HPV protein. For example, “HIV-1 Vpr induces cell-cycle arrest with expression of HPV-16 E6”

Response: We have removed all mention of “unchecked tumor growth and survival” to instead provide more detailed outcomes of HIV-1 protein and HPV interactions.

  1. In table 2 and 3: Providing the unknown mechanism with unknown HIV-1 protein does not make any sense.

Response: We removed the final row of each of the tables.

Reviewer 2 Report

Many of the previous comments are addressed.

However what is still lacking is the interaction of ART affecting the other viruses, for example, there are tons of literature showing RTIs impacting HBV titre, and just to name one https://doi.org/10.1093/abt/tbac005 that discussed conserved targeting in viral proteins with one specific druggable RT site doi: 10.3390/molecules25245902 against HIV that may include other viruses such as HBV titers https://doi.org/10.1159/000360942 and

doi: 10.1128/AAC.00599-13, https://doi.org/10.3892/etm.2020.9467 among many others. Given the nature of the review, such ART effects on other viruses should be discussed at least briefly in the respective viral sections e.g HPV doi: 10.1186/1750-9378-5-8 , will leave the authors to google.

In fact, in the vein of the cross-interactions of the immune system, the authors may cite that immune responses against HIV proteins

Table 3: Unknown, Unknown with immunological altercations is fairly meaningless. HIV is known to cause immune altercations, so to have unknown unknown and stating the obvious should perhaps best be removed.

It is my opinion that the addition of the above will add the value to the review for the readers.

Author Response

Reviewer #2:

  1. Many of the previous comments are addressed. However, what is still lacking is the interaction of ART affecting the other viruses, for example, there are tons of literature showing RTIs impacting HBV titre, and just to name one https://doi.org/10.1093/abt/tbac005 that discussed conserved targeting in viral proteins with one specific druggable RT site doi: 10.3390/molecules25245902 against HIV that may include other viruses such as HBV titers https://doi.org/10.1159/000360942 and doi: 10.1128/AAC.00599-13, https://doi.org/10.3892/etm.2020.9467 among many others. Given the nature of the review, such ART effects on other viruses should be discussed at least briefly in the respective viral sections e.g HPV doi: 10.1186/1750-9378-5-8 , will leave the authors to google.

Response: We appreciate this question and have added short discussions of how cART may affect other viruses. The following sections were added to the HBV/HCV and HPV co-infection sections, respectively.  The discussion was also expanded.

“Therapeutic interventions for HIV-1/HBV co-infections rely on reverse transcriptases with structurally similar enzymatic pockets and thus are susceptible to the same enzymatic inhibitors [214].  The Food and Drug Administration approved four nucleoside reverse transcriptase inhibitors (NRTI) that target HIV, i.e. lamivudine, emtricitabine, tenofovir disoproxil fumarate, and tenofovir alafenamide, for use in HBV. While these NRTI impeding HBV polymerase activity and viral replication [215], they demand potentially life-long regimens and are not curative.  Accordingly, clevudine-triphosphate, which shares functionalities with both NRTI and non-nucleoside reverse transcriptase inhibitors, has been introduced to address the weaknesses of NRTI [216]. In HIV-1/HCV co-infections, HIV targeted cART is ineffective at reducing HCV replication.  Instead pan-genotypic direct-acting antivirals (i.e., NS5A inhibitors and NS3/4A protease inhibitors) in are used to cure most persons of HCV infection in 12 – 24 weeks depending on state of liver fibrosis [217].  However, ART and HCV treatment regimens must be selected with special consideration for potential drug-drug interactions and overlapping toxicities as many combinations are counter indicated [218].  Direct-acting antivirals are a dramatic improvement over previous interferon-based therapies that had significant adverse effects and counterindications for PLWH [217,219,220].  In both HIV-1/HBV and HCV co-infections, the benefits of cART outweigh the risks of co-therapies.  However, the need for anti-co-infection therapeutics capable of targeting oncogenic viruses like HBV/HCV without detrimental effects on HIV suppression or drug induced toxicity.”

“The intersection of cART on HPV is difficult to define, perhaps because of the heterogeneity and ubiquity of HPV infection in the world population. A recent study demonstrated that amprenavir, a protease inhibitor discontinued in 2007, compromised cell-cell junctions in a stratified model of oral or cervical epithelium. In this model, protease inhibitors increased the permeability of HPV16 into the basal layer to promote infection and viral spread [258]. However, the effect of amprenavir on HPV progression in PLWH has not been examined. Many studies have not found an association between cART, HIV and HPV incidence, prevalence, or progression [259-261]. However, an HIV Epidemiology Research Study found an increased HPV clearance rate in HIV positive women diagnosed with squamous intraepithelial lesions on cART compared to HIV negative women [262]. Further, a meta-analysis of cross-sectional, cohort studies of women living with HIV between 1996 and 2017 revealed that early cART initiation and sustained adherence reduces incidence and progression of HPV-associated lesions [263]. Regardless, there is no cure for HPV and the interplay between HIV-1 infection, cART and HPV progression remains unclear. Thus, to effectively suppress dual infections, it is imperative that pharmaceuticals which simultaneously target the both HIV-1 and HPV are developed or that cART must be safely co-administered with anti-oncovirus therapies.”

  1. In fact, in the vein of the cross-interactions of the immune system, the authors may cite that immune responses against HIV proteins. Table 3: Unknown, Unknown with immunological altercations is fairly meaningless. HIV is known to cause immune altercations, so to have unknown unknown and stating the obvious should perhaps best be removed.

Response: We removed the final row of each of the tables.

  1. It is my opinion that the addition of the above will add the value to the review for the readers.

Response: We greatly appreciate your valuable feedback and opportunity to enhance our manuscript.